# INarIG: Iterative Non-autoregressive Instruct Generation Model For Word-Level Auto Completion

**Hengchao Shang, Zongyao Li, Daimeng Wei, Jiaxin Guo,**
**Minghan Wang, Xiaoyu Chen, Lizhi Lei, Hao Yang**
Huawei Translation Service Center, Beijing, China
{shanghengchao,lizongyao,weidaimeng,guojiaxin1,
wangminghan,chenxiaoyu35,leilizhi,yanghao30}@huawei.com

## Abstract

Computer-aided translation (CAT) aims to enhance human translation efficiency and is still important in scenarios where machine translation cannot meet quality requirements. One fundamental task within this field is Word-Level Auto Completion (WLAC). WLAC predicts a target word given a source sentence, translation context, and a human typed character sequence. Previous works either employ word classification models to exploit contextual information from both sides of the target word or directly disregarded the dependencies from the right-side context. Furthermore, the key information, i.e. human typed sequences, is only used as prefix constraints in the decoding module. In this paper, we propose the INarIG (Iterative Non-autoregressive Instruct Generation) model, which constructs the human typed sequence into Instruction Unit and employs iterative decoding with subwords to fully utilize input information given in the task. Our model is more competent in dealing with low-frequency words (core scenario of this task), and achieves state-of-the-art results on the WMT22 and benchmark datasets, with a maximum increase of over 10% prediction accuracy.

## 1 Introduction

Transformer and its variants (Vaswani et al., 2017; Shaw et al., 2018; So et al., 2019; Dehghani et al., 2018), coupled with large-scale parallel and synthetic corpora (Sennrich et al., 2016; Edunov et al., 2018; Wu et al., 2019; Pham et al., 2020), have boosted machine translation quality to a large extent. Now, Machine Translation (MT) can basically meet users' requirements in scenarios where translation quality is undemanding, for instance, understanding social media posts. However, in other scenarios where audience expect highly accurate translations (e.g. reading a product manual or a government document), manual proofreading and post-editing are still required. Re-

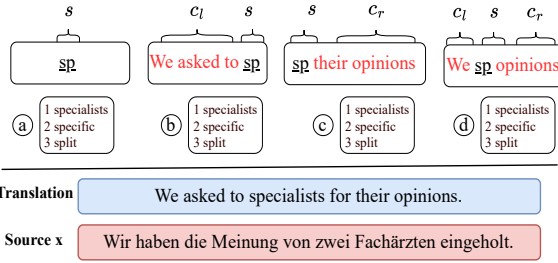

Figure 1: Illustration of WLAC task. $c = (c_l, c_r)$ is the translation context for source sentence $x$. $s$ is the human typed sequence. Words in the rounded rectangles are predictions generated by the model given the tuple $(x, c, s)$. As $c_l, c_r$ can be empty, the task has four types of context data: (a) zero-context, (b) prefix, (c) suffix, and (d) bi-context.

search on computer-aided translation (CAT) (Barrachina et al., 2009; Green et al., 2014a; Knowles and Koehn, 2016; Santy et al., 2019) aims at enhancing manual translation. Among these works, sentence/word-level autocompletion (Knowles and Koehn, 2016; Zhao et al., 2020; Li et al., 2021) is a fundamental task and has been widely applied to Post Editing (PE) and Interactive Translation Prediction (ITP) (Knowles and Koehn, 2016).

Li et al. (2021) offer a more general definition of Word-Level AutoCompletion (WLAC) (Figure 1): predicting a target word $w$ given the source sentence $x$, translation context $c$ (left and right to the target word), and human typed sequence $s$ (one or several character(s) of the target word). And they open-source the first benchmark system with training datasets and baseline results.

According to the definition, the task faces two challenges: First, the target word may have contextual dependencies on both left and right sides. Consequently, during decoding, it is imperative to consider information from both sides. Second, the human typed sequence is merely a prefix sequence of the target word with an uncertain length. Thus, how to fully utilize this information is a key to the

task.

In prior works (Navarro et al., 2022; Moslem et al., 2022), dependencies from the right side are disregarded during decoding, and only information from the left side is utilized. Additionally, the human typed sequence is merely used as a prefix constraint in the decoding module. As a result, it is difficult to achieve further performance enhancement based on these methods. Some other works (Li et al., 2021; Yang et al., 2022) use word-level models to perform word classification. These methods are capable of integrating information dependencies from both sides, but word-level models are incapable of predicting low-frequency and out-of-vocabulary (OOV) words, thus cannot satisfy translators' major needs as they type low-frequency words the most (Casacuberta et al., 2022).

In this paper, we propose a new Iterative Non-autoregressive Instruct Generation (INarIG) model for WLAC. First, we construct the human typed sequence $s$ into an Instruction Unit to instruct model generation and encode it together with the translation context to fully leverage available information. As human typed sequence is merely one or several characters, we perform character-level embedding. Second, we use conditional masked decoding, which is similar to Non-autoregressive Translation (NAT) model, to ensure compatibility with dependencies from both side of the decoding anchor (target word). This decoding strategy enables one model to process four types of context. Moreover, we employ iterative decoding of sub-words to form the target word, which makes the model more friendly to low-frequency words and thus more appealing to translators.

Another major challenge in this task, which was ignored in previous research, is the incomplete translation text. As $c_r$ and $c_l$ only provide fragmented information, modeling the target language is challenging, which in turn affects the model performance. We propose two strategies to enhance the model's language modelling performance: (1) fine-tuning on a pre-trained MT model or Conditional Masked Language Model (CMLM) (Ghazvininejad et al., 2019); and (2) multi-task learning with the CMLM task.

Our models have achieved state-of-the-art performance, with prediction accuracies that exceed the benchmarks by an average of 7.76%, and a maximum increase of over 10%.

Our main contributions include:

- Constructing the human typed sequence into Instruction Unit and performing character-level embedding to ensure deep information fusion at the encoding phase.

- Iterative decoding at subword level ensures compatibility with context dependencies on both sides, making the model more competent in handling low-frequency words.

- Utilizing pre-training and multi-task learning strategies to efficiently address incomplete translation context.

## 2 Background

### 2.1 Task Definition

According to Li et al. (2021), the WLAC task illustration is shown in Figure 1, and a detailed definition is as follows: Suppose $x = (x_1, x_2, ..., x_m)$ is a source sequence and $s = (s_1, s_2, ..., s_k)$ is a sequence of human typed characters. The translation context is denoted as $c = (c_l, c_r)$, where $c_l = (c_{l,1}, c_{l,2}, ..., c_{l,i}), c_r = (c_{r,1}, c_{r,2}, ..., c_{r,j})$. The translation pieces $c_l$ and $c_r$ are on the left and right hand side of $s$, respectively. WLAC aims to predict a target word $w$, which starts with $s$ and is to be placed in the middle of $c_l$ and $c_r$ to constitute a complete translation. Note that $w$ is not necessary consecutive to $c_{l,i}$ or $c_{r,1}$. More generally, $c_r$, $c_l$ can be empty, which lead to four types of context: zero-context (no context), suffix (only right context), prefix (only left context), and bi-context (left and right context).

### 2.2 Main Challenges

As a new task, WLAC presents unique challenges for modeling both its input and constraints. These challenges include:

- The human typed sequence contains prefix information of the target word, but its length is uncertain, which poses a challenge for its utilization.

- The target word may have contextual dependencies to both the left and right sides. As a result, the decoding module must be adapted to simultaneously utilize information from both sides.

- Translation context ($c_r$ and $c_l$) are only information segments and may not be contiguous with the target word. Incomplete information on the translation side poses a significant challenge for modeling the target language.

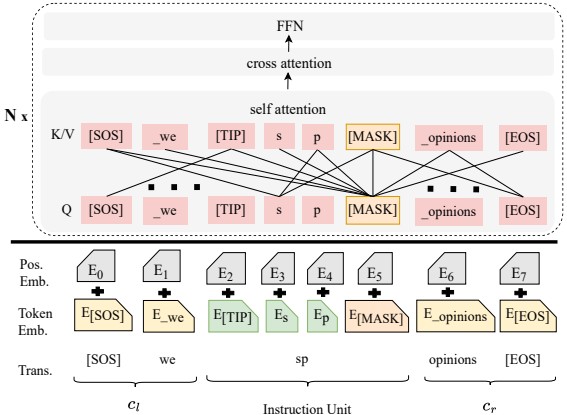 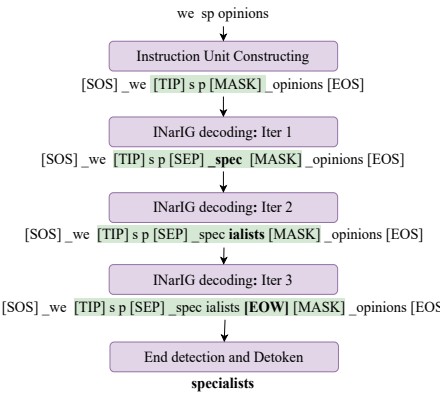

Figure 2: The figure on the left illustrates the decoder structure of our model. The human typed sequence "sp" is constructed to Instruction Unit with character-level embedding. On the right, we provide an example of our iterative decoding process. [TIP] and [SEP] are special tokens used in Instruction Unit. The [SEP] token is only present when decoded subwords exist. The [EOW] token is used to mark the end of a word.

- In languages such as Chinese where the writing and phonetic symbols are inconsistent, the translation-side language model should be able to simultaneously process Chinese characters, Pinyin, and their relationship.

## 2.3 Related Work

Prior to a general definition of this task provided by Li et al. (2021), related researches include interactive MT (IMT) (Green et al. (2014b), Santy et al. (2019)) or translation prediction (TP) (Koehn (2009), Alabau et al. (2014), Koehn et al. (2014), Green et al. (2014b), Huang et al. (2015), Knowles and Koehn (2016) and Coppers et al. (2018)), Langlais et al. (2000), Santy et al. (2019), Lee et al. (2021). Theses works focus on predicting the next word/phrase given the left-side translation context. The settings of the task share great similarity with WLAC, so these works have referential importance to WLAC.

Navarro et al. (2022) and Moslem et al. (2022) adapt the strategy mentioned above to this task. Unfortunately, the models do not fully fit the WLAC task setting. According to their model designs, some inputs $(s, c_r)$ or constraints ($w$ is not necessary consecutive to $c_l$ or $c_r$) are not leveraged.

Li et al. (2021) and Yang et al. (2022) treat WLAC as a word classification task using a word-level model. The human typed sequence is used as a vocabulary constraint during decoding. Such models are well-suited for the WLAC task setting, but it does not make full use of human typed sequence and is not very friendly to low-frequency and out-of-vocabulary words. Ailem et al. (2022)

adopt the most similar approach as ours: use a subword-level model for autoregressive decoding. The human typed sequence is used as a soft constraint and the target word is generated by the decoder. However, they do not particularly address the incomplete translation, which hinders the model performance.

## 3 Methodology

In this section, we first present the overall structure of our model. Then, we introduce the Instruction Unit we used, as well as our decoding procedure. Figure 2 shows the details.

### 3.1 Conditional Masked Decoding

Our model's overall structure is a Conditional Masked Language Model (Ghazvininejad et al., 2019), which is similar to that in the benchmark system (Li et al., 2021). The source-side information is encoded by the encoder, and its representation is passed to the decoder by cross attention. A [MASK] token serves as decoding anchor and translation context $(c_r, c_l)$ is integrated as inputs to be passed to the decoder. When decoding with [MASK] token, the model can utilize contextual information from both sides and also allows us to process four types of context using a single model. We denote $D_{wlac} = D_{zero} \cup D_{pre} \cup D_{suf} \cup D_{bi}$ as the training data for WLAC, and try to minimize the following objective:

$$L(D_{wlac}; \theta) = \sum_{(x,c,s) \in D_{wlac}} \log P(w|x, c, s; \theta)$$

(1)

## 3.2 Instruction Unit

Human typed sequence, as the prefix of the target word, is a key to the WLAC task. In order to simplify the model and ensure maximum utilization of information, we choose to directly encode it into the model. For this purpose, we construct the sequence into an Instruction Unit using a special token [TIP] and integrate it directly into the decoder's input together with translation context. Additionally, the human typed sequence is merely a prefix sequence of uncertain length so character-level embedding is used. Figure 2 shows the details. In section 6.2, we demonstrate that the Instruction Unit can guide the model to generate the correct target word.

## 3.3 Iterative Decoding with Subword

To better adapt to the task setting, we choose to use subword as the encoding unit. Consequently, in order to predict a complete target word, we need to generate a set of subwords. Inspired by the non-autoregressive model, we introduce iterative decoding, which decodes one subword per iteration. Then the decoded subword is also integrated into the Instruction Unit to guide the generation of the next subword. This process repeats until the final word is formed. The decoding process is carried out from left to right, as the human typed sequence serves as the prefix of the target word.

An example is shown in the right part of Figure 2. The [EOW] token is used to mark the end of a word, and the [SEP] token serves as a separator between the human type sequence and decoded subword tokens.

## 4 Training Strategy

### 4.1 Pre-training & Multi-Task learning

As mentioned in section 2.2, $c_l$ and $c_r$ are randomly simulated incomplete translation pieces, posing great challenges to train the target-side language model (LM). To address this, we propose the following two strategies.

First, inheriting the target-side LM ability from a pre-trained model. Related pre-training tasks include MT and CMLM.

Another solution is multi-task joint training. By incorporating an additional task, we can ensure that the model can maintain language learning capability throughout training. CMLM can be an additional task trained together with WLAC as it uses

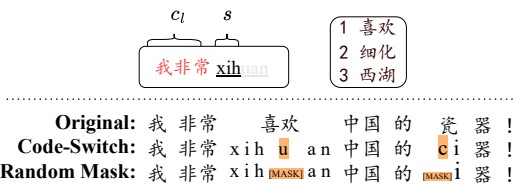

Figure 3: The upper part shows an example of Chinese translation. The Chinese words need to predicted with the human typed sequence "xih", which is a prefix of the corresponding Pinyin "xihuan". The lower part demonstrates Code-Switch and Random Mask for Chinese sentence.

the same data structure. The overall optimization objective is:

$$
L(\theta) = \sum_{(x,c,s) \in D_{wlac}} \log P(w|x,c,s;\theta) \\
+ \sum_{(x,c') \in D_{cmlm}} \log P(w_{mask}|x,c';\theta)
\tag{2}
$$

Note that the $D_{cmlm}$ is the CMLM data generated from bilingual data. $c'$ is the complete translation context in CMLM data.

### 4.2 Code-Switching for Chinese

Another typical issue regarding Chinese is the conversion between Chinese characters and Pinyin (phonetic symbols). Most people need to input the Pinyin first and then use an input method editor to convert it into Chinese words, see the upper part of Figure 3. Since Pinyin is not used in pre-training or multi-task learning, the model has no knowledge of the structure of Pinyin and the mapping between Chinese words and Pinyin, which poses another challenge to this task.

To address this issue, we refer to the code-switching strategy and convert a portion of Chinese words into Pinyin sequence when constructing the CMLM training data, thus the model is able to learn the mapping between Pinyin and Chinese characters. We also mask character(s) in Pinyin sequence with a certain probability, allowing the model to learn the inner structure of Pinyin. A case is shown in the lower part of Figure 3.

## 5 Experiment

### 5.1 Datasets

To validate our model's effectiveness, we perform experiments on two datasets: one from

WMT22[1] and the other from the Benchmark system. Each dataset includes data of two language pairs (Chinese-English and English-German) on both directions. Regarding Chinese↔English data, the benchmark dataset contains 1.25M bilingual sentence pairs from LDC corpora while the WMT22 dataset contains 15M pairs from UN Parallel Corpus V1.0[2]. pypinyin[3] is used to convert Chinese word to phonetic symbols. For the English↔German tasks, the two datasets use the same data: 4.5M from WMT14 and is pre-processed by Stanford[4]. We first use the data generation script[5] provided by the WMT22 task to construct training data $D_{wlac}$ from bilingual pairs. Both datasets contain their corresponding test sets. We adopt sentencepiece (Kudo and Richardson, 2018) for subword modeling and set the vocabulary size to 32K.

## 5.2 Model Configuration

We adopt the Transformer-base configuration for the purpose of fair comparison. The batch size is set to 64K tokens with a learning rate of 5e-4 and a 4000 step warmup. Other parameters are set to the default values of fairseq (Ott et al., 2019) [6]. All of our experiments are performed on NVIDIA 8*V100 and we save a model every 2K updates.

## 5.3 Training Process

We train a single model to process all types of translation context. The model leverages the joint training strategy mentioned in §3.1. During training, we divide the training process into three phases according to the two strategies presented in 4.1:

1) Generate the final training data $D_{train}$ from $D_{wlac}$ according to 3.3.

2) Pre-train an MT or CMLM model with the original bilingual pairs.

3) Finetune it with $D_{cmlm}$ and $D_{train}$ using multi-task joint training.

Models are measured by loss on dev sets. Particularly, for CMLM task pre-training, we set the probability of mask token to 15%-50%, to ensure the model learn target-side LM. In the multi-task learning phase, we set the ratio of $D_{cmlm}$ and $D_{wlac}$

data as 1:1, and the probability of mask token in CMLM data is 20%.

Code-switching for the EN⇒ZH model is performed during the multi-task training phase. We randomly select 50% of the $D_{cmlm}$ data for code switching. Each Chinese token has a 30% chance of being selected and converted to Pinyin characters.

## 5.4 Inference & Evaluation

During inference, we average the last 10 checkpoints to obtain the final model and report results from a single model without ensemble. We adopt beam search during decoding and the beam size is set to 4 for models of all language pairs.

We use accuracy as the evaluation metric (Li et al., 2021):

$$ACC = \frac{N_{match}}{N_{all}} \qquad (3)$$

where $N_{match}$ is the number of correctly predicted words and $N_{all}$ is the number of all test examples.

## 6 Results

### 6.1 Main Results

We first compare our method with other related works on the WMT22 test sets in Table 1. The results demonstrate that our model achieves higher accuracy (an average increase of 4.8% and a maximum increase of 7.58%) compared with the best results from other works, resulting in a new state-of-the-art performance. Additionally, we also compare our model with the benchmark system (Li et al., 2021) on the benchmark test sets. Results are shown in Table 2. And the results remain consistent with that of WMT22 dataset, with an average accuracy improvement of over 7% and a maximum improvement of over 10%. The results fully validate the effectiveness and robustness of our method.

### 6.2 Effectiveness of Instruction Unit

One of our primary contributions is constructing the human typed sequence into character-level Instruction Unit, which efficiently encodes prefix constraint information into the model to guide word generation. To demonstrate its effectiveness, we conduct a comparative experiment by replacing the Instruction Unit with a vocabulary filter module similar to the one used in the benchmark system. Results are presented in Table 3.

[1]https://www.statmt.org/wmt22/word-autocompletion.html

[2]https://conferences.unite.un.org/uncorpus
[3]https://github.com/mozillazg/python-pinyin
[4]https://nlp.stanford.edu/projects/nmt/
[5]https://github.com/lemaoliu/WLAC
[6]https://github.com/facebookresearch/fairseq

| Models | ZH⇒EN | EN⇒ZH | DE⇒EN | EN⇒DE |
|---|---|---|---|---|
| Li et al. (2021) (our reproduction) | 52.06 | 50.86 | 60.13 | 56.18 |
| Moslem et al. (2022) | 50.41 | 31.94 | 61.44 | 58.94 |
| Ailem et al. (2022) | - | - | 57.36 | 48.97 |
| Yang et al. (2022) | 54.05 | 53.98 | 57.27 | 41.83 |
| Navarro et al. (2022) | - | - | 39.02 | 33.97 |
| INarIG (Ours) | **59.10** | **56.23** | **69.02** | **63.52** |

Table 1: Experiment results on WMT22 dataset and "-" means not provided. The results of benchmark system (Li et al., 2021) are reproduced by us.

| Models | ZH⇒EN | | EN⇒ZH | | DE⇒EN | | EN⇒DE | |
|---|---|---|---|---|---|---|---|---|
| | NIST05 | NIST06 | NIST05 | NIST06 | NT13 | NT14 | NT13 | NT14 |
| Li et al. (2021) | 55.54 | 55.85 | 53.64 | 54.25 | 57.84 | 56.75 | 56.91 | 52.68 |
| INarIG (Ours) | **62.50** | **64.16** | **59.31** | **60.76** | **65.21** | **65.76** | **61.22** | **62.76** |

Table 2: Experiment results on benchmark dataset. NIST05 and NIST06 are test sets used in the benchmark system for Chinese-English (both directions), and test sets for German-English tasks are newstest13 and newstest14.

| Models | EN⇒ZH | | DE⇒EN | |
|---|---|---|---|---|
| | NIST05 | NIST06 | NT13 | NT14 |
| INarIG | 59.31 | 60.76 | 65.21 | 65.76 |
| w/o IU | 53.60 | 53.85 | 58.64 | 57.71 |
| w/o Iter-D | 55.61 | 56.82 | 60.45 | 61.22 |

Table 3: Results of the comparative experiments based on benchmark dataset. "w/o IU" means replace Instruction Unit with a vocabulary filter module. "w/o Iter-D" means replace subword based iterative decoding with a word-level model.

According to the results, Instruction Unit achieves consistent enhancement across various language pairs. Particularly, in terms of accuracy, we observe 6.6% to 8% increase on DE⇒EN and 5.7% to 6.9% increase on EN⇒ZH. The results indicate that model-based deep information fusion is much more effective than shallow fusion in the decoding module.

## 6.3 Effectiveness of Iterative Decoding

To improve performance with low-frequency words, we utilize a subword-level model and an iterative decoding strategy. To verify the effectiveness of this strategy, we train a word-level model to perform word classification and use it for comparison. We compare the accuracies on benchmark test set, as shown in Table 3. The accuracies of our models outperform the word-level model, with an average accuracy improvement of around 4.2%, verifying the effectiveness of our decoding strategy.

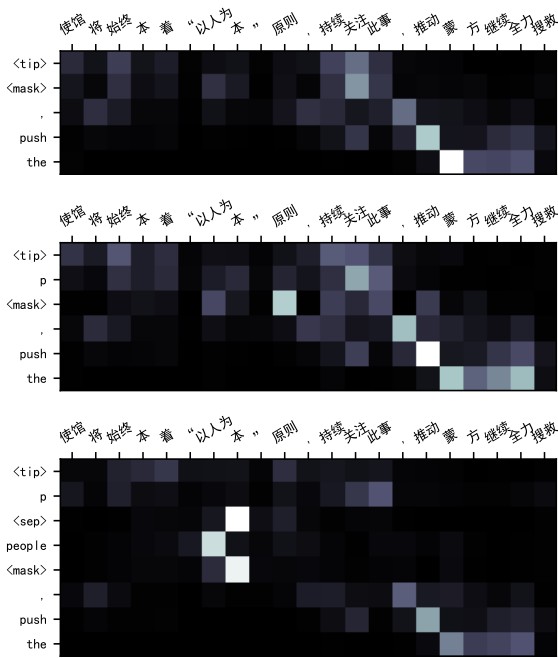

Figure 4: A case study of cross attention-based word alignment for [MASK] token with source sentence.

## 7 Anlysis

In this chapter, we analyzes the effectiveness of our innovative points and optimization strategies used in training to better validate our method. Unless otherwise specified, the experiments are conducted on the benchmark dataset.

## 7.1 Instruct Generation

We analyze the Instruct Generation method used in our model from the perspective of word align-

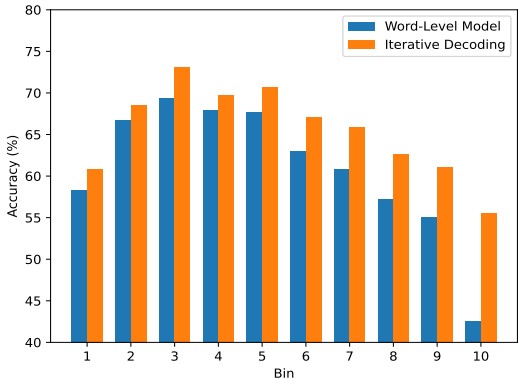

Figure 5: The accuracies of words at different frequencies for DE⇒EN task. The test sets newstest13 and newstest14 are merged and divided to 10 bins with equal size according to the frequency of target word. Bin 1 and Bin 10 denote the most frequent and infrequent bin.

| Models | EN⇒ZH | | DE⇒EN | |
|---|---|---|---|---|
| | NIST05 | NIST06 | NT13 | NT14 |
| Raw-Model | 50.69 | 52.34 | 53.21 | 51.82 |
| +NMT-FT | 56.24 | 58.33 | 63.66 | 64.12 |
| +CMLM-FT | 56.44 | 57.25 | 63.58 | 63.80 |
| +Multi-Task | 57.28 | 59.41 | 64.00 | 64.75 |
| INarIG | **59.31** | **60.76** | **65.21** | **65.76** |

Table 4: The results of our experiments on different training strategies. Raw-Model refers to the baseline model trained directly on WLAC data. NMT-FT and CMLM-FT uses NMT and CMLM pre-trained models, respectively. Multi-Task means multi-task training with both CMLM and WLAC data. The INarIG uses a combination of an NMT pre-trained model and multi-task training.

ment. According to the task setting, the translation context is incomplete, making it difficult for the [MASK] token to align with the corresponding information in the source text. The prefix information of the target word in Instruction Unit enhances the representation of the [MASK], facilitating alignment and guiding decoding. We use an example to visualize the last cross-attention layer's attention weights, as shown in Figure 4.

In this case, the candidate words are "principle" and "people-oriented" with the corresponding source tokens "原则" and "以人为本". When comparing the top and middle figures, the addition of prefix "p" increases the attention weights for the correct original token, especially for "原则". In addition, if subword "people" has already be decoded, the target word should only be "people-centered". As shown in the bottom figure, the model shift the attention to focus on "本" in the source side.

The case confirms that our Instruction Unit can effectively guide the model to generate correct words, improving the accuracy of the model.

## 7.2 Accuracy & Word Frequency

As mentioned before, in real-world scenarios, translators have a stronger demand for autocompletions of low-frequency words (Casacuberta et al., 2022). In Section 6.3, we have already demonstrated the effectiveness of our subword-level model in terms of overall accuracy. Here, we conduct further analysis on the accuracy of words at different word frequencies for additional validation.

The results for DE⇒EN task are shown in Figure 5, which demonstrate that the subword-level model

is superior at all frequency intervals, especially for low-frequency words. For 30% less-frequent words, the accuracy improves by more than 8%. The results of the EN⇒ZH task shown in appendix §A, also exhibit almost the same trend, confirming that our model has an advantage in predicting low-frequency words.

## 7.3 Pre-training & Multi-Task learning

As stated above, we realize that incomplete translation context challenges the model performance. To prove the effectiveness of our pre-training and multi-task strategy, we conduct experiments on the benchmark dataset, and the results are shown in Table 4.

According to the results, the accuracy of our models drops 7% to 13% when no related strategy is applied. To be more specific, we observe pre-training on either NMT or CMLM task achieves equivalent results and multi-task training leads to greater improvements than pre-training. The final model which combines the two strategies, obtains further 1.2% improvement on average.

Furthermore, our analysis of the training loss curve is consistent with the findings. In addition, we evaluate the performance of our models on four types of context and observe consistent improvements, demonstrating the robustness of our model. Details of this analysis is presented in the appendix §B.

## 7.4 Code-Switching for Chinese

In order to verify the effectiveness of code-switching strategy for Chinese, we conduct experiments on EN⇒ZH using both datasets, and the

| Models | EN⇒ZH | | |
| --- | --- | --- | --- |
| | NIST05 | NIST06 | WMT22 |
| INarIG | 59.31 | 60.76 | 56.23 |
| w/o CS | 58.02 | 59.77 | 55.07 |

Table 5: Results of the code-switching strategy comparative experiments for Chinese. "w/o CS" means without code-switch strategy during training.

| Models | EN⇒ZH | | DE⇒EN | |
| --- | --- | --- | --- | --- |
| | NIST05 | NIST06 | NT13 | NT14 |
| INarIG | 59.31 | 60.76 | 65.21 | 65.76 |
| + BT | 60.66 | 61.83 | 67.89 | 68.12 |
| +Big | 61.56 | 63.75 | 69.97 | 70.80 |

Table 6: Results of BT style synthetic data on benchmark dataset. "+ BT" means back translation style synthetic data is added for training, and "+Big" means changing model configuration to transformer Big.

accuracy results are summarized in Table 5.

The code-switching strategy can increase the accuracy of the model by an additional 1-2%, which also confirms the necessity of modeling the internal structure of Pinyin and the relationship between pinyin and Chinese words. It is worth noting that Chinese is a representative example of this type of language, and the code-switching strategy can be easily extended to similar languages.

## 7.5 Enhancement with Monolingual data

Due to the high degree of similarity between our model and the standard MT models, we are able to transfer existing optimization strategies from MT tasks. Back translation (Sennrich et al., 2016; Edunov et al., 2018; Wu et al., 2019) is the simplest and most effective one. To confirm that back translation can be applied to our model, we conduct an additional experiment by adding 10M Chinese and 20M English monolingual data on top of the Benchmark dataset. The beam-search back translation (Sennrich et al., 2016) is used to construct synthetic data. The results are shown in Table 6.

Overall, the BT-style synthetic data can improve the accuracy of the model, and increasing the model capacity can further enhance the model performance. This demonstrates that our model can effectively adopt optimization strategies from other NLP tasks, such as MT, and it is worth further research.

## 7.6 Inference Performance

Theoretically, our model has higher computational complexity compared to word-based classification models. Regarding the iterative decoding process of a non-autoregressive model, the representations of all tokens on the decoder side need to be computed repeatedly. However the impact on inference speed is not serious in practical applications for two reasons: (1) The task predicts only one word so the sequence length is limited. (2) Computing can be processed in parallel, and shallow-decoder (Kasai et al., 2020; Wang et al., 2019) Transformer models are widely used.

## 8 Discussion & Future Work

Two issues requires more research and analysis:

1. Our model allows us to apply more mainstream enhancement strategies in the field of natural language processing, such as multilingual enhancement and word alignment. How to efficiently use these strategies should be further explored.

2. Our decoding strategy is inspired by non-autoregressive models. More general decoding strategies that integrate autoregressive and non-autoregressive approaches have been studied by Wang et al. (2022); Li et al. (2022), and future researches may find better strategies.

## 8.1 LLM for WLAC

Upon the release of ChatGPT[7], we observe that large language models exhibit exceptional performance across a range of NLP tasks. To evaluate ChatGPT's capability on WLAC, we extract a subset of 1000 instances from the EN⇒DE newstest14 test set and construct a corresponding task prompt (provided in the appendix §C).

The accuracy on our test set is 44.2%. ChatGPT's performance on the WLAC task is inferior to our method (63.5% in terms of accuracy). Improving prompts may improve its efficacy. Furthermore, large language models suffer from slow inference speeds, and the word completion task requires minimal latency to ensure seamless input. Overall, further exploration and adaptation may be needed for the application of large models to the WLAC task.

## 9 Conclusion

In this paper, we propose a novel iterative non-autoregressive instruct generation model for the

---

[7]https://chat.openai.com/

WLAC task, and validate the effectiveness of our model on two mainstream datasets. Subsequently, through carefully-designed experiments, we verify that our model's better capability of utilizing avaialbe information compared to previous works, as our approach uses model-based deep information fusion. We also demonstrate that our iterative decoding based on subwords can improve the accuracy of low-frequency words and better meet translators' requirements. Afterwards, through comparative experiments, we confirm that a series of training strategies used in our work help improve accuracy.

## 10 Limitations

The iterative NAT decoding strategy we used can decode one or several tokens in a single iteration, depending on the number of masks being input. We choose to decode one token per iteration. Further research should be done to explore the effectiveness of decoding more than one tokens per iteration. In addition, NAT decoding can be performed from left to right, right to left, or randomly. We employ the most straightforward approach: left to right. Further research is required to analyze the performance of other decoding directions.

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

## A  Accuracy & Word Frequency for EN⇒ZH task

Accuracy results of words with different frequencies for EN⇒ZH task is presented in Figure 6.

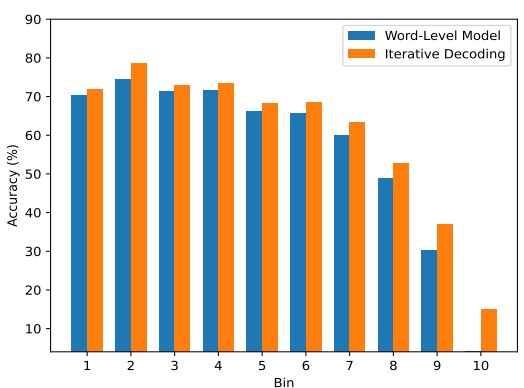

Figure 6: The accuracies of words at different frequencies for EN⇒ZH task. The test sets NIST05 and NIST06 are merged and divided to 10 bins with equal size according to the frequency of target word. Bin 1 and Bin 10 denote the most frequent and infrequent bin.

## B  Pre-training vs Multi-Task learning

Taking DE⇒EN as an example, we visualize loss curves of our models on the dev set, as shown in Figure 7. The model without any strategy gets stuck in a local optima in few updates, causing loss increase in subsequent training. After we add CMLM data (complete translations) for multi-task training, the lowest loss drops from 4.2 to 3.5. The loss reduces as well for the model using a pre-training strategy. However, we observe loss increase after 250K update steps, probably due to catastrophic forgetting during fine-tuning. The LM ability learned in the pre-training stage deteriorates in the fine-tuning stage, so the lowest loss of the model is still higher than modal using the multi-task training strategy. The combination of NMT-FT and multi-task strategies ensures stable training and leads to the lowest loss.

Finally, we visualize the accuracy of each model under different types of context in the right side of Figure 7. Two conclusions can be drawn: (1) The accuracy is basically consistent with the loss performance, which again verifies our analysis. (2) The accuracy curves under different types of context are consistent too, which proves generalizability of our optimization strategy. We also perform the same ablation experiment on the EN⇒ZH model, and the results are consistent with those of the DE⇒EN model. The detailed results are in Figure 8.

## C  Prompt for WLAC

The specific prompt structure can be found in Figure 9.

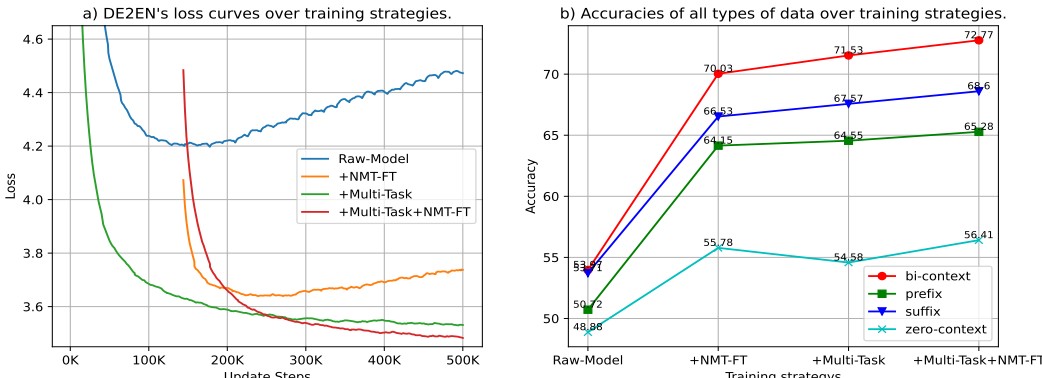

Figure 7: The loss curves and accuracies for DE⇒EN models with different training strategies. All results are measured on the NT13 dev test.

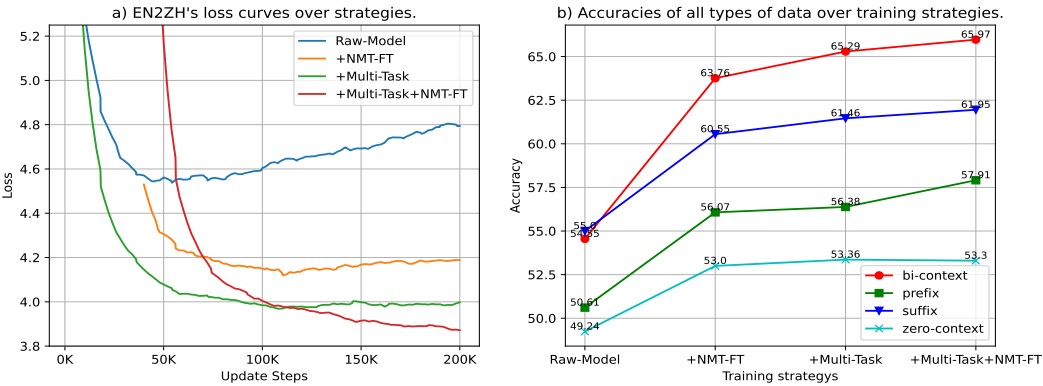

Figure 8: The loss curves and accuracies for EN⇒ZH over training strategies. All results are based on the NIST02 dev test.

---

**Q:** Word-Level AutoCompletion (WLAC), which aims to predict a target word given a source sentence, translation context and a human typed character sequence. To make the task more general in real-world scenarios, the translation context is made that the left context and right context, which can be empty. The original text is English, the translation is German.
Source sentence is "***If you can do that , khan said , the public gets more comfortable .***".
left context is "***damit , so khan , wäre auch***".
right context is "***.***".
human typed character sequence is "***be***".
So what's the word that starts with "***be***"? (It's best to give the answer without explanation.)

**A:** The word that starts with "***be***" is "***bereit***".

---

Figure 9: The example of ChatGPT prompt about the WLAC task.