# OpenReview forum: "INarIG: Iterative Non-autoregressive Instruct Generation Model For Word-Level Auto Completion"
_EMNLP/2023/Conference — EMNLP 2023 Findings_

### Official Review · Reviewer_Tsoo · 2023-08-04

**Soundness:** 3

**Excitement:**

4: Strong: This paper deepens the understanding of some phenomenon or lowers the barriers to an existing research direction.

**Paper Topic And Main Contributions:**

This work propose a new model for the task of word auto completion in assisted translation. They use an iterative non-autoregressive decoder that is trained to predict characters as conditional masked LM (one subword at the time) until completing the word. In addition, they use pre-training and multi-task learning strategies to boost the task performance. The evaluation results on datasets performed on 2 languages pairs (in both directions) show large improvement over previous methods.

**Questions For The Authors:**

- What is the theoretical complexity of the method in comparison with previous approaches?

**Reasons To Accept:**

- The method is well described and the main design decisions are justified. The improvements over previous methods are large, which shows the effectiveness of the proposed approach.

- The experiments are sound, they show ablations over different components and a variety of analysis that covers the most relevant questions. This includes: how the model works with low frequency words, different pre-training strategies, additional monolingual data, and comparison with ChatGPT.


**Reasons To Reject:**

- The main application of this task is probably on-line, thus latency of the model’s inference should be a factor to consider. Predicting one subword at the time requires running the model N times (N being the number of subword). Also, having a list of suggested candidates increases the latency by C times (C being the number of candidates). The paper does not include latency analysis.

**Reproducibility:**

4: Could mostly reproduce the results, but there may be some variation because of sample variance or minor variations in their interpretation of the protocol or method.

**Reviewer Confidence:**

4: Quite sure. I tried to check the important points carefully. It's unlikely, though conceivable, that I missed something that should affect my ratings.

---

> ### Author Rebuttal · Authors · 2023-08-29
>
> **Q:** What is the theoretical complexity of the method in comparison with previous approaches?
>
> **A:**   Thanks for your thoughtful review.
>
> Theoretically, our model has higher computational complexity compared to that of word-based classification models. Regarding the iterative decoding process of a NAT model, the representations of all tokens on the decoder side need to be computed repeatedly. However, the inference of the NAT model decoder can be fully parallelized (including generating multiple candidate sets), and the number of iterations N is limited (average 2-3), so the inference latency of the model is acceptable in practical application.
>
> In our experiment, the latency to predict one word is within 50ms with a Tesla T4 GPU powered by pytorch. Further optimizations (adapting acceleration libraries: FasterTransformer [1] or LightSeq [2]) can reduce the inference latency to within 15ms. Even for on-line systems, this level of performance is sufficient.
>
>
>
> [1]  https://github.com/NVIDIA/FasterTransformer
>
> [2] Wang et al., 2021 LightSeq: A high performance inference library for transformers. Association for Computational Linguistics.

---

### Official Review · Reviewer_4ENL · 2023-08-04

**Soundness:** 2

**Excitement:**

2: Mediocre: This paper makes marginal contributions (vs non-contemporaneous work), so I would rather not see it in the conference.

**Paper Topic And Main Contributions:**

The concept of Word-Level AutoCompletion (WLAC) involves predicting a desired word when provided with a source sentence, translation context, and at least one character input from a human. This research introduces the Iterative Non-autoregressive Instruct Generation (INarIG) model, which systematically produces subwords to form the final predicted word. Additionally, the proposed approach refines its performance by fine-tuning a pre-trained MT Conditional Masked Language Model (CMLM) to address incomplete translation context. Notably, the integration of character embedding is employed to incorporate human-typed sequence information into the model effectively.

**Reasons To Accept:**

The paper presents an innovative Iterative Non-autoregressive Instruction Generation (INarIG) model, which effectively handles intricacies in Word-Level AutoCompletion (WLAC) through subword-level decoding. The model is fine-tuned on a pre-trained MT Conditional Masked Language Model (CMLM).

The INarIG model demonstrates a significant improvement in accuracy compared to current methods and benchmark systems. This advancement raises the bar in WLAC.


**Reasons To Reject:**

In contrast to Li et al. (2021), which predicts target words at the word level, the proposed model iteratively generates a subword to compose the prediction word. The proposed method also fine-tunes on a pre-trained MT Conditional Masked Language Model(CMLM) to enhance the model since the translation text is incomplete. Similar to this proposed model, Ailem et al. (2022) also use a subword-level model for autoregressive decoding to generate a sequence of subwords to constitute a target word step by step. So, the major difference between Ailem et al. (2022) and the proposed model is finetuning from a pre-trained CMLM. Convincing validation would require a comparative analysis of both models (the proposed model and Ailem et al. 2022), either both with or without fine-tuning.

The paper predominantly focuses on a specific task (Word-Level AutoCompletion), potentially lacking broader applicability to other NLP domains, which could limit its significance to a diverse conference audience.


**Reproducibility:**

3: Could reproduce the results with some difficulty. The settings of parameters are underspecified or subjectively determined; the training/evaluation data are not widely available.

**Reviewer Confidence:**

3: Pretty sure, but there's a chance I missed something. Although I have a good feel for this area in general, I did not carefully check the paper's details, e.g., the math, experimental design, or novelty.

---

> ### Author Rebuttal · Authors · 2023-08-29
>
> Thanks for your thoughtful review.
>
> Our model shares some similarities with Ailem et al. (2022), but achieves 11-14% higher accuracy on the same data and model size. We use no additional training data for pretraining and multitask learning, and use no other pre-training models as well.
>
> The key to our significant improvement is that we identify the adverse impact of incomplete translations on model training, and propose a novel solution accordingly. We design a model that is highly compatible with the standard MT task and address such adverse impact using pre-training (MT models) and multitask learning (WLAC and CMLM). We believe this is one of the major contributions of our work, with detailed descriptions in Sections 1, 4.1 and 7.3. Ailem et al. (2022) do not analyze and propose solutions in this respect.
>
> In addition, the model architecture we employed is different from that in Ailem et al. (2022). In order to accommodate context dependencies of the right-side target words, we choose to use an NAT model, so as to put all translation information at the decoder side. This design ensures seamless compatibility with MT tasks and allows us to easily transfer optimization strategies in the MT field (e.g. pre-training, multitask learning, back translation augmentation). On the contrary, Ailem et al. (2022) encode both source and target context at the encoder side, and the decoder only generates target words. Their model design is significantly different from a standard MT task setting, so it requires customized optimization strategies, making optimization hard to be performed.
>
> In all, our contributions to the WLAC task include:
> 1. Analyze the characteristics and challenges of the WLAC task;
> 2. Design a feasible and efficient model tailored for the WLAC task;
> 3. Propose a training strategy to address major challenges of the task and achieve SOTA results with significant improvement;
>
> Thank you again for your questions.

---

### Official Review · Reviewer_DbUp · 2023-08-05

**Soundness:** 3

**Excitement:**

3: Ambivalent: It has merits (e.g., it reports state-of-the-art results, the idea is nice), but there are key weaknesses (e.g., it describes incremental work), and it can significantly benefit from another round of revision. However, I won't object to accepting it if my co-reviewers champion it.

**Paper Topic And Main Contributions:**

This paper presents a new neural architecture called INarIG for word-level autocompletion in computer-aided translation. Word autocompletion aims to predict a target word given a source sentence, translation context, and a human-typed prefix. Specifically, it designs an Instruction Unit to fully encode the human prefix using characters, which provides more direct guidance to the model's generation process. The model uses an iterative, non-autoregressive decoder with subwords for compatibility with contextual dependencies on both sides of the target word. This also handles low-frequency words better. The model employs pre-training and multi-task learning to address the challenge of incomplete context. For Chinese, it uses a code-switching strategy to learn mappings between characters and pinyin. The experiments demonstrate state-of-the-art results on two standard datasets, with especially strong improvements on low-frequency words. The paper provides extensive analysis and ablation studies to validate the advantages of the proposed techniques.

**Questions For The Authors:**

1. Regarding the training data analysis without using the code-switching strategy in Section 7.4: Specifically, how are these data composed? To be more precise, what would the user's input sequence look like if this strategy were not employed? Concerning the experimental results in Section 7.4, which indicate that the impact of the code-switching strategy is only 1-2%: Why is this influence so minimal?

2. The right panel in Figure 2 and the [MASK] symbol in Figure 3 are not the same, and there are distinct differences in their functionalities. In Figure 2, [MASK] is used as a separator, and it is not intended for predicting words or characters. However, in Figure 3, [MASK] serves as a placeholder that the model needs to predict, whether it is a word or a character. Is this difference expected?

**Reasons To Accept:**

1. Propose INarIG to fully utilize the human-typed sequences in the WLAC task.

2. The model achieves new state-of-the-art results on two standard datasets, demonstrating its effectiveness.


**Reasons To Reject:**

1. Limited novelty. Several model components like the NAT decoder and code-switching have been applied in other contexts before. The innovations seem more incremental to existing ideas.

2. The results regarding the decoding speed are missing.

**Reproducibility:**

3: Could reproduce the results with some difficulty. The settings of parameters are underspecified or subjectively determined; the training/evaluation data are not widely available.

**Reviewer Confidence:**

4: Quite sure. I tried to check the important points carefully. It's unlikely, though conceivable, that I missed something that should affect my ratings.

---

> ### Author Rebuttal · Authors · 2023-08-29
>
> Thank you for your thoughtful review.
>
> ### For question 1：
>
> Regarding the code-switching strategy used in the EN-ZH model, let us explain it using the example shown in Figure 3 in our paper.
>
> In this example, the Chinese words used are "喜欢" (like) and "瓷" (porcelain). Their corresponding pinyin is "xihuan" and "ci".
>
> Here is a table summarizing the different types of data used during model training:
>
> | Data type       | Source                              | Target                                         | mask token |
> | :-------------- | :---------------------------------- | ---------------------------------------------- | ---------- |
> | $D_{bilingual}$ | I like Chinese porcelain very much. | 我 非常  喜欢 中国  的 瓷  器 ！               | -          |
> | $D_{wlac}$      | I like Chinese porcelain very much. | 我   [TIP] x i h [MASK]  瓷  器 ！             | 喜欢       |
> | $D_{cmlm}$      | I like Chinese porcelain very much. | 我 非常  [MASK] 中国  的 [MASK] 器 ！          | 喜欢  瓷   |
> | $D_{cmlm+ cs}$  | I like Chinese porcelain very much. | 我 非常 x i [MASK] u a n 中国  的 [MASK] 器 ！ | h  瓷      |
>
> **Q:**  Regarding the training data analysis without using the code-switching strategy in Section 7.4: Specifically, how are these data composed?
>
> **A:** The code-switching strategy is performed during the multi-task learning stage, when the model needs to learn $D_{wlac}$ and $D_{cmlm}$ (both generated from the original $D_{bilingual}$) simultaneously.
>
> At this time, the correspondence between pinyin and Chinese word only appears in the $D_{wlac}$ , and the pinyin characters "xih" entered by the user are only a prefix of the complete pinyin string"xihuan".
>
> The code-switching strategy converts Chinese words in the $D_{cmlm}$ into corresponding pinyin with a certain probability, and the pinyin characters also participate in random masking. Then we get a new dataset denoted as $D_{cmlm+ cs}$ for training.  $D_{cmlm+ cs}$ allows the model to learn the pinyin structure and correspondence between pinyin and Chinese words too.
>
> **Q:** To be more precise, what would the user's input sequence look like if this strategy were not employed?
>
> **A:**  From the previous examples, we can see that whether the code-switching strategy is used or not, the user's input is a prefix (e.g. "xi", "xih") of the pinyin string ("xihuan") corresponding to the target Chinese word ("喜欢").
>
> Details for this example are shown in the table below:
>
> | Information type    | Information content                             |
> | ------------------- | ----------------------------------------------- |
> | Source              | I like Chinese porcelain very much.             |
> | Translation context | left context: 我         right context: 瓷 器 ! |
> | Human-input         | xih                                             |
> | Decoder input       | 我 [TIP] x i h [MASK] 瓷 器 !                   |
> | Target word         | 喜欢                                            |
>
> **Q:** Concerning the experimental results in Section 7.4, which indicate that the impact of the code-switching strategy is only 1-2%: Why is this influence so minimal?
>
> **A:**  Regarding the effect of code-switching strategy, we speculate the reason is: When there is sufficient simulated $D_{wlac}$ generated, the model has already established a strong baseline, so the supplementary pinyin information in $D_{cmlm+ cs}$  only lead to minor improvement. This is worth further study.
>
> ### For question 2：
>
> **Q:** The right panel in Figure 2 and the [MASK] symbol in Figure 3 are not the same, and there are distinct differences in their functionalities. In Figure 2, [MASK] is used as a separator, and it is not intended for predicting words or characters. However, in Figure 3, [MASK] serves as a placeholder that the model needs to predict, whether it is a word or a character. Is this difference expected?
>
> **A:** Regarding the role of [MASK], our paper might not explain it clearly enough. The [MASK] in both Figure 2 and Figure 3 serves as a placeholder for decoding.
>
> First, the right side of Figure 2 describes the iterative decoding process of the model, where each round of iterative decoding is based on the [MASK]. Figure 3 mainly shows the process of constructing training data for the CMLM task during multi-task training. Here [MASK] also serves as a placeholder used to predict the target word.
>
> ### About decoding speed:
>
> Our model has higher computational cost during inference compared to word-based classification models, because the representations in the decoder need to be recomputed at each iteration. However, since the current task only predicts one word, the number of iterations is limited. Moreover, computations at the decoder side during inference can be fully parallelized, so the overall inference latency does not have a significant increase. In our experiment, the latency to predict one word is within 50ms with a Tesla T4 GPU powered by pytorch. Further optimizations (adapting acceleration libraries: FasterTransformer [1] or LightSeq [2]) can reduce the inference latency to within 15ms. Even for online systems, this level of performance is sufficient.
>
> [1]  https://github.com/NVIDIA/FasterTransformer
>
> [2] Wang et al., 2021 LightSeq: A high performance inference library for transformers. Association for Computational Linguistics.

---

### Meta-Review · Area_Chair_6YzA · 2023-09-18

**Recommendation:** 3

**Metareview:**

This paper tackles the word-level auto-completion task (WLAC), useful for computer-assisted translation. The authors propose a new method that, unlike previous approaches, can use target right-context in additional to left context, and handles the fact that either left and/or right context may be empty or incomplete. They experiment with pretraining and co-training on the MLM task along with WLAC to account for the fact that language modelling is done with incomplete target context. They also integrate an experiment with code-switched Pinyin-Chinese character data. They achieve better performance than the benchmark model, perform ablation studies on the model components and perform better on low-frequency words, which are the most important in the task.

The method surpasses the previous state-of-the-art results and although it builds on previous work, introduces a couple of novel elements (character-level encoding for the target prefix, pretraining and multi-task fine-tuning, iterative decoding process with an non-autoregressive MLM model). Ablations are included and the experiments appear sound. It is also important that the method improves results for low-frequency words. The main missing detail is the decoding speed and complexity, which would be important for real-world CAT applications. The authors provide some details about this in their rebuttal. This discussion should be added to the paper and would benefit from a comparison with other methods.

---

### Decision · Program_Chairs · 2023-10-07

**Decision:**

Accept-Findings

**Comment:**

This paper tackles the word-level auto-completion task (WLAC), useful for computer-assisted translation. The authors propose a new method that, unlike previous approaches, can use target right-context in additional to left context, and handles the fact that either left and/or right context may be empty or incomplete. They experiment with pretraining and co-training on the MLM task along with WLAC to account for the fact that language modelling is done with incomplete target context. They also integrate an experiment with code-switched Pinyin-Chinese character data. They achieve better performance than the benchmark model, perform ablation studies on the model components and perform better on low-frequency words, which are the most important in the task.

The method surpasses the previous state-of-the-art results and although it builds on previous work, introduces a couple of novel elements (character-level encoding for the target prefix, pretraining and multi-task fine-tuning, iterative decoding process with an non-autoregressive MLM model). Ablations are included and the experiments appear sound. It is also important that the method improves results for low-frequency words. The main missing detail is the decoding speed and complexity, which would be important for real-world CAT applications. The authors provide some details about this in their rebuttal. This discussion should be added to the paper and would benefit from a comparison with other methods.